# Spatial and Temporal Variations of Habitat Quality and Its Response of Landscape Dynamic in the Three Gorges Reservoir Area, China

**DOI:** 10.3390/ijerph19063594

**Published:** 2022-03-17

**Authors:** Shuangshuang Liu, Qipeng Liao, Mingzhu Xiao, Dengyue Zhao, Chunbo Huang

**Affiliations:** 1Key Laboratory of Regional Ecology and Environmental Change, School of Geography and Information Engineering, China University of Geosciences, Wuhan 430074, China; liushuangshuang@cug.edu.cn (S.L.); zhaodengyue@cug.edu.cn (D.Z.); 2State Key Laboratory of Biogeology and Environmental Geology, School of Geography and Information Engineering, China University of Geosciences, Wuhan 430074, China; 3School of Arts and Communication, China University of Geosciences, Wuhan 430074, China; liaoqp@cug.edu.cn (Q.L.); xiao_mingzhu@cug.edu.cn (M.X.)

**Keywords:** biodiversity, ecological restoration, habitat quality, InVEST model, land use change

## Abstract

Habitat quality is an important indicator for assessing biodiversity and is critical to ecosystem processes. With urban development and construction in developing countries, habitat quality is increasingly influenced by landscape pattern changes. This has made habitat conservation to be an increasingly urgent issue. Despite the growing interest in this issue, studies that reveal the role of land use change in habitat degradation at multiple scales are still lacking. Therefore, we analyzed the spatial and temporal variations of habitat quality of the Three Gorges Reservoir area by the InVEST habitat quality model and demonstrated the responses of habitat quality to various landscape dynamics by correspondence analysis. The result showed that the habitat quality score of this area increased from 0.685 in 2000 to 0.739 in 2015 and presented a significant spatial heterogeneity. Habitat quality was significantly higher in the northeastern and southwestern parts of the reservoir area than in other regions. Meanwhile, habitat quality improved with altitude and slope, and increased for all altitude and slope zones. The habitat quality of >1000 m and >25° zone exceeds 0.8, while the habitat quality of <500 m and <15° zone is less than 0.6. Habitat quality significantly varied among landscape dynamics and was extremely sensitive to vegetation recovery and urban expansion. The vegetation restoration model of returning farmland to forest is difficult to sustain, so we suggest changing the vegetation recovery model to constructing complex vegetation community. This study helps us to better understand the effects of landscape pattern changes on habitat quality and can provide a scientific basis for formulating regional ecological conservation policies and sustainable use of land resources.

## 1. Introduction

Habitat quality is defined as the resources and conditions present in an area that are capable of producing occupancy (including survival and reproduction) by a particular organism [1]. In general, habitat quality varies with the intensity of nearby land use. Biodiversity is intimately linked to the production of ecosystem services [2]. It is spatial in nature and can be estimated by analyzing land use maps in conjunction with threats to the species’ habitat [3,4]. Habitat quality and rarity can be used as proxies for biodiversity, ultimately estimating the range of habitats and vegetation types across the landscape, as well as their degradation status [5]. So, habitat quality is critical to the change process of ecosystems and can accurately reflect the level of regional ecological services and ecological security [6,7,8,9]. In recent years, the habitat value has been widely recognized from both social and ecological perspectives, as habitat is a key element for the continued survival of species and can provide multiple ecosystem services to humans [4]. Usually, high-quality habitats are relatively complete, with higher stability in ecosystem structure and function, and its ecosystem could recover quickly after disturbance [10,11]. However, habitat quality is vulnerable to multiple factors, such as land conditions, surrounding natural environment, social economy, and so on [12,13]. With urban development and construction in developing countries, habitat conservation is increasingly threatened by anthropogenic factors, particularly the landscape pattern changes that are seen as key factors in habitat quality degradation [9]. Landscape pattern changes can fundamentally alter ecosystem composition and configuration, and ultimately affect energy flow and material cycling among habitat patches [9]. Therefore, habitat conservation while still meeting the needs of urban development and construction has become an increasingly urgent issue [4].

Changes in landscape pattern will lead to corresponding changes in the composition of the ecosystem as well as biodiversity. The change in habitat quality can directly exhibit the regional change in biodiversity and landscape pattern [14]. With the emergence of a series of ecological and environmental problems at the landscape and species levels, the study of landscape pattern evolution and habitat quality relationships caused by land use change provides a solution to the problem of ecological security [15]. Since the 1990s, many scholars have conducted research in this aspect, hoping to provide a scientific basis for analyzing ecological changes, formulating regional ecological conservation policies, and sustainable use of land resources [9]. For example, Bai et al. [15] analyzed the spatial and temporal characteristics of landscape patterns and habitat quality in Changchun City using a combination of spatial analysis and ecological models, and further explored the spatial heterogeneity of the effects of urbanization on habitat quality. Xu et al. [16] quantified the spatial and temporal evolution of land use, landscape pattern and habitat quality in the Taihu basin from 1985 to 2015 based on ArcGIS software and the InVEST model. Berta Aneseyee et al. [17] analyzed the habitat quality changes from 1988 to 2018 by taking the Winike Watershed in the Omo-Gibe Basin, Southwest Ethiopia as the study area, and they also discussed the correlation between the influencing factors and habitat quality in this region. Although many scholars have explained the effects of landscape patterns on habitat quality changes, there remains a dearth of studies that reveal the role of land use change in habitat degradation [8]. Therefore, it is necessary to conduct studies on the spatial and temporal evolution of habitat quality, to explore the effects of landscape evolution on habitat quality changes, and to enhance the maintenance and management of regional ecosystem stability.

Field experiments and ecological model simulations are important tools for assessing habitat quality [18,19]. Field studies mainly evaluate habitat quality through various biodiversity indices, such as species richness, the Simpson index of diversity, the Shannon–Weaver index, and so on. For instance, Yudharta et al. [20] documented that the Shannon–Weaver biodiversity index of orthopteran decreases with altitude. Wu et al. [21] used species richness, Shannon-Wiener index, and Simpson index to analyze the spatial patterns of species diversity in arboreal communities of typical Mori ecosystems in China. However, these studies usually focus on small spatial scales and the results are likely to be affected by the site conditions and monotonous organisms [19]. Therefore, many ecological models were used for habitat quality assessment and conservation, such as Artificial Intelligence for Ecosystem Services (ARIES), Social Values for Ecosystem Services (SoIVES), and Integrated Valuation of Ecosystem Services and Tradeoffs (InVEST) [9]. ARIES has been used for spatial mapping/quantification of services and valuation of services, however, it not good at measuring temporal changes [22]. SolVES is a tool for mapping and analyzing social survey response data [23]. Respondents’ subjective factors have a greater influence on the assessment of habitat quality. In comparison, the InVEST model developed by Stanford University and the WWF has been widely used for its accurate quantification, visualization of results, ease of operation and high data processing power [24]. The habitat quality module of the InVEST model establishes the link between suitability and threats of different land covers, and then assesses the distribution and degradation of habitat quality in different landscape patterns based on the sensitivity of different habitats to threat sources [9]. Applying this module to assess the impacts of landscape evolution and ecological conservation policies on terrestrial habitats can effectively maintain biodiversity and safeguard ecosystem health, helping to guide landscape management and decision-making at a regional scale.

As one of the biodiversity hotspots in China [25], the Three Gorges reservoir (TGR) area is experiencing significant environmental changes, especially the increased anthropogenic pressure due to the construction of the Three Gorges Dam and urban expansion, as well as various national policy changes. With the storing water of the reservoir, animals in the area below the submerged zone were forced up the mountains, which destroyed the ecological balance of the migrating site. Meanwhile, the invasion of non-indigenous species may accelerate the extinction of some rare species. For example, the Chinese river dolphin (Lipotes vexillifer) has been endangered by the Gezhouba Dam which is 38 km downstream from the TGD. In addition, the construction of the TGD has turned dozens of hilltops into land-bridge islands, which presented a fragmented habitat. Human activities such as land-use changes, hydrological disturbance, non-point source pollution, over-exploitation and exotic species can accelerate species extinction [26]. Therefore, the biodiversity and ecosystem functions of the TGR area have to adjust to the newly created landscape configuration and regional context [24]. To balance biodiversity conservation and the sustainable socio-economic development of the reservoir area, a variety of ecological restoration projects have been implemented. For example, the National Forest Protection Program carried out some activities such as afforestation, re-vegetation to improve the ecological environment of the reservoir area [19,27]. These ecological restoration projects could effectively mitigate the negative impact of the TGD construction on the biodiversity of the reservoir area and strengthen the protection of species’ habitats and promote the sustainability of species’ habitats [28]. However, these projects have a broad scale and long-lasting duration, and their impact on terrestrial ecosystem biodiversity and restoration remains unclear. This means that there is a huge potential threat to current habitat conservation and ecological restoration in the TGR area, which is of great research value. Therefore, the TGR area can be a good case study for exploring and quantifying the impact of landscape pattern evolution on habitat quality in ecologically fragile areas.

In this paper, we used the InVEST model to evaluate the habitat quality and its spatial-temporal evolutions of terrestrial ecosystems in the TRG area. The main purpose is to reveal the biodiversity and ecosystem health of the reservoir area under ecological restoration activities and provide a scientific basis for the subsequent ecological restoration planning and implementation. Based on this research motives, our study objectives are as follows: (1) analyzing the spatial and temporal variations of the habitat quality in the TGR area; (2) demonstrating the characteristics in habitat quality of various landscape types and their response to the landscape configuration evolution; (3) proposing land use policies to promote the habitat quality of terrestrial ecosystem in the TGR area.

## 2. Materials and Methods

### 2.1. The Study Area

The TGR area is located in the upper and middle reaches of the Yangtze River, and lies between 28°31′−31°44′ N and 105°50′−111°40′ E (Figure 1). It refers to 20 counties of Hubei Province and Chongqing that have been affected by the construction of the TGD. The reservoir area includes 16 districts and counties of Chongqing (Wuxi, Wushan, Fengjie, Yunyang, Kaixian, Wanzhou, Zhongxian, Shizhu, Fengdu, Wulong, Fuling, Changshou, Yubei District, Banan District, Chongqing City and Jiangjin City) and 4 counties in southwestern Hubei (Yiling, Zigui, Xingshan and Badong), and its total area is approximately 5.8 × 10^4^ km^2^. The TGR area belongs to the humid monsoon climate in the mid-subtropical zone, with abundant water and heat resources. The mean annual temperature is 20 °C and the mean annual precipitation is 1200 mm [29]. The terrain of the entire reservoir area is complex, and the highest altitude exceeds 2500 m. In this region, mountains account for about 74%, hills account for about 21.7%, and plains account for about 4.3%. Located at the transition area between eastern and western China, the per capita arable land in the TGR area is small, the soil erosion is serious and the land suitable for urban construction is limited due to a vast mountainous area and hilly terraces, which also leads to a fragile ecosystem [30].

As a typical biodiversity hotspot [25,31], the TGR area has about 6388 species of higher plants, 523 species of terrestrial vertebrates, 3418 species of insects, 350 species of fish, and 1085 species of plankton, benthic and aquatic plants [32,33]. Meanwhile, the TGR area is rich in plant resources with about 208 families, 1428 genera and 6088 species of vascular plants [32]. Six major vegetation types exist in the TRG area, including coniferous, broad-leaved, mixed broad-leaved, mixed coniferous and broad-leaved forests, bamboo forests and shrubs. The coniferous forest is mainly distributed in areas at altitudes above 1700 m, and the mixed forest is mainly located in areas at altitudes of 1300–1700 m. The hills below 1300 m or less mainly are broad-leaved forests and shrubs. Human activities such as the construction of the TGD and land use changes have brought about a series of negative impacts on ecosystem health and biodiversity in the reservoir area [25]. The implementation of some ecological projects has mitigated the degradation of habitat quality in the reservoir area [33], but their impacts on biodiversity and ecosystem restoration remain unclear.

### 2.2. Evaluation Model of Habitat Quality

In this study, the InVEST habitat quality model [5] was adopted to evaluate the habitat quality of terrestrial ecosystems in the TGR area. This model treats habitat quality as a continuous variable ranging from low to medium to high based on the resources available for survival, reproduction, and population persistence, respectively [1]. Habitat quality scores were used as a proxy for biodiversity to estimate the habitat quality of each landscape and their degradation status [5,18]. This model assumes that the legal protection of land is effective and that all threats to a landscape are additive [5]. It generates habitat quality assessment maps by combining information on habitat suitability and threats to biodiversity, which allows rapid identification of changes in habitat quality and quantity [34,35]. It assumes that the grid with higher habitat quality indicates a richness of native species, while the grid with lower habitat quality indicates the opposite. We calculated the habitat quality by combining landscape sensitivity and external threat intensity and documented the spatial and temporal variations of habitat quality to characterized biodiversity change.

In this model, the land use type was considered as the legal protection of land. However, the patch isolation caused by fragmented landscapes reduces the habitat suitability. Since the data on specific biodiversity-habitat relationships are lacking, we took a simple binary approach to assign habitat to the land use types. We used a relative habitat score ranging from 0 to 1 to replace the simple binary approach, which could distinguish the habitat quality differences among various land use types. The habitat suitability is positively correlated with the habitat score. Habitat quality is decided by the following four factors: each threat’s relative impact, the distance between habitats and sources of threats, the degree to which the land is legally protected, and the relative sensitivity of each habitat type to each threat.

(1) The relative impact of each threat

The relative impact of each threat on habitat quality was determined to represent the different damage potentials of each threat. Some threats may be more damaging to habitat quality when other conditions are equal, indicating a higher relative impact of the threats. Therefore, we set the weights at higher levels for the destructive threats to reflect the relative impact of the threats [18]. The weights were range from 0 to 1, and the higher score means stronger destructiveness of threats.

(2) The distance between habitat and the impact of threats on habitat

The impact of a threat on habitat decreases as the distance from the degradation source increases, so that grid cells that are more proximate to threats will experience higher impacts [5]. Therefore, the impact of a threat on habitat partly depends on how quickly they decrease or decay, over space. The linear distance-decay function (Equation (1) and the exponential distance-decay function (Equation (2) were used to describe the impact of a threat on habitat decreases.
(1)$$\;i_{rxy}=1-\left(\frac{d_{xy}}{d_{r,max}}\right),$$
(2)$$i_{rxy}=\exp\left(-\left(\frac{2.99}{d_{r,max}}\right)d_{xy}\right)$$
where, i*_rxy_* denotes the impact of threat r that originates from *y* on habitat *x*; d*_xy_* is the linear distance between habitat *x* and *y*; d*_r,max_* is the maximum effective distance of threat *r*.

(3) The degree to which the land is legally protected

A formally protected area can effectively mitigate the effects of threats to the habitat. The model assumes that the more legal protection from degradation a habitat has, the less it will be affected by nearby threats. Protection from legal/institutional/social/physical sources affects the accessibility of habitat in the study area. Thus, an accessible coefficient between 0 and 1 is used in the model to represent the degree that habitats are effect by threats, where 0 indicates complete inaccessibility while 1 indicates complete accessibility, which means there is no protection in the region [5,34].

(4) The relative sensitivity of each habitat type to each threat

The relative sensitivity of each habitat type to each threat on the landscape is the final factor used when generating the total degradation in a cell with habitat. The model assumes that the more sensitive a habitat type is to a threat, the more degraded the habitat type will be by that threat. A sensitivity coefficient ranging from 0 to 1 was used to reflect the relative sensitivity of each habitat type to each threat, where values closer to 1 indicate greater sensitivity [5].

Therefore, the total degradation level in each pixel is given by the equation below (Equation (3)), the habitat quality is negatively related with the degradation score.
(3)$$D_{xj}=\overset{R}{\underset{r=1}{\sum }}\overset{Y_{r}}{\underset{y=1}{\sum }}\left(\frac{w_{r}}{\sum_{r=1}^{R}w_{r}}\right)r_{y}i_{rxy}p_{x}s_{jr},$$
where, *D_xj_* denotes the degradation of habitat type *j* on habitat *x*; *R* is the total number of threat factors; *r* denotes threat factor *r*; *Y_r_* indicates the set of grid cells on *r*’s raster map; *w_r_* denotes the weight of the relative extent of the threat factor *r*; *r_y_* denotes threat resource *r* on habitat *y*; *i_rxy_* indicates the impact of threat *r* that originates from *x* on habitat *y*; *p_x_* denotes accessibility of habitat *x*; *s_jr_* is a sensibility score that habitat type *j* response to threat factor *r*.

After calculating the degradation score for habitat type by combining all of the threat factors, it is translated into a habitat quality value using a half saturation function (Equation (4)). As a grid cell’s degradation score increases its habitat quality decreases.
(4)$$Q_{xj}=H_{j}\left(1-\frac{D_{xj}^{z}}{D_{xj}^{z}+k^{z}}\right)$$
where, *Q_xj_* is the final habitat quality score of habitat type *j* on habitat *x*; *H_j_* is the habitat suitability of habitat type *j*; *D_xj_* denotes the degradation of habitat type *j* on habitat *x*; *z* and *k* are scaling parameters (or constants).

### 2.3. Data Sources and the Parameter Settings

The study data include the land use data, DEM (Digital Elevation Model) data, road data and points of interest (Table 1). The land use map has nine land use types, including coniferous forest, broadleaf forest, mixed forest, shrub, grassland, cropland, water, built-up land, and bare land [5]. It was derived from our previous studies [29]. Meanwhile, we downloaded DEM data with 30-m resolution from ASTER Global Digital Elevation Model V002 (http://www.gscloud.cn/, accessed on 13 November 2018) to obtain elevations and generate slope maps using ArcGIS spatial analysis tool (version 10.2.2). The road data were downloaded from the National Geomatics Center of China (http://ngcc.sbsm.gov.cn/, accessed on 13 November 2018), mainly including railway, highway, and national road. In addition, some point data such as tourist attractions, hotels as well as traffic stations were acquired from Baidu maps.

The input data of the InVEST habitat quality model include the land-use maps, the threats data, the half-saturation constant, the habitat types, and the sensitivity of habitat types to each threat.

(1) Land-use maps

A total of 4 land use maps in 2000, 2005, 2010 and 2015 were used to identify the variations of habitats for different periods. A relative habitat suitability score (*Hj* in the Equation (4)) was assigned to each land use type ranging from 0 to 1 where 1 indicates the highest habitat suitability. A ranking of less than 1 indicates habitat where a species or functional group may have lower survivability.

(2) The threats data

We selected 9 threat factors, including 3 linear vector data, 3-point vector data and 3 binary raster data (Table 2). In the linear vector data, the maximum distance of railway and national road was 1 km, with a weight of 0.5 and 0.8, respectively. Additionally, the highway data has the same weight as the national road data. The maximum distance could be referred to d_r,max_ in Equation (2) and the weight be mentioned as *w_r_*, which means a weighted value of the relative extent of the threat factor in Equation (3). In the point vector data, the maximum distance and weight value of traffic station, hotel and tourist attraction were sequentially reduced. Their maximum distances were 10 km, 5 km, 3 km, respectively; their weight values were 1, 0.7 and 0.6, respectively. Furthermore, the raster data included build-up area, water body and cropland data. The threat from build-up area on habitat decreased in an exponential trend, while the threats from water body and cropland decreased in a linear trend. The weight of build-up area, water and cropland data were 1, 0.3 and 0.5, respectively. The build-up area, water and cropland were derived from land use map. In general, the influence degree of line threat source is using exponential attenuation model (Equation (2)) while that of point threat source is using linear model (Equation (1)).

(3) Habitat scores for different landscapes and the sensitivity of landscapes to each threat

In this study, we assumed that forestland had the highest habitat score, while grassland and cropland had lower scores. Habitat scores of different landscapes and their sensitivity to threat factors were obtained by experts, the specific scores were shown in Table 3. The habitat score for each landscape means a relative habitat suitability score (*Hj* in the Equation (4). A relative habitat score ranging from 0 to 1 where 1 indicates the highest habitat suitability. A ranking of less than 1 indicates habitat where a species or functional group may have lower survivability. In this study, coniferous forest, broadleaf forest, mixed forest, and shrub exhibited the highest habitat suitability, with a habitat score of 1. The habitat score of grassland and cropland were 0.8 and 0.5, respectively. Moreover, the habitat score of water, built-up land and bare land were 0 because these landscapes are not suitable to be a habitat. Besides, the responses to threat factors of land use types produced a relative sensitivity score, *s_jr_* in Equation (3). All threats should be measured in the same scale and units (i.e., all measured in density terms, or all measured in presence/absence terms) and not through some combination of metrics. In the table, the column name is the threat factor, while the row name is the habitat type (land use type).

### 2.4. Statistic Analysis

#### 2.4.1. Trend Analysis

Trend analysis was used to evaluate the rate or direction of changes in a variable over the period. In our study, this method was used to identify the changing trend of habitat quality. Refer to the relevant studies [36], we chose the least squares linear regression model (Equation (5)) to analyze and describe the changing trend of habitat quality in the TGR area, and finally passed the t-test of the slope and the *p*-value to reflect the significant characteristics of these variations.
Y = ax + b(5)
where, y is the habitat quality indicator; and x denotes the time (year). The a is the modeled slope, which could reflect the annual variation rate of the habitat quality indicators. And the b denotes the intercept of the regression model.

#### 2.4.2. Correspondence Analysis

Correspondence analysis was used to consider distances between research objects by combining classification methods with graphical modeling [37,38]. To determine the relationship between landscape change features and habitat quality change, we counted the number of grids with habitat quality change features at a 1 km grid scale based on spatial information statistics. Correspondence analysis first discretizes the number of grids and then puts the data into a contingency table. The column names contain the two pollutant change characteristics and their names, while row names contain the names of the landscape change [31]. We used the correspondence analysis function in R language and the distance between the row and column categories embedded in the correspondence analysis graph to explore the relationship between the landscape change characteristics and the habitat quality change characteristics.

#### 2.4.3. Hierarchical Clustering Analysis (HCA)

Hierarchical Clustering Analysis (HCA) is a technique used to find the underlying structure or clustering tendency of objects through an iterative process that associates (agglomerative methods) or dissociates (divisive methods) the objects based on the information contained in the fingerprint matrix [39]. In this paper, the hierarchical agglomerative clustering method was used to cluster the habitat quality scores of different counties according to the similarity of the objects. Compared to other clustering methods, the agglomerative methods provide structured clustering with valuable information on the levels of similarity and relative distance between clusters [40]. Moreover, the Euclidean distance was also used to calculate the similarity value between the habitat quality of each county, which was used as a criterion to construct the clustering tree.

## 3. Results

### 3.1. Habitat Quality of the TGR Area

Based on the InVEST model, we evaluated the habitat quality of the TGR area between 2000 and 2015 (Figure 2a). The habitat quality score of the TGR area increased from 0.685 in 2000 to 0.739 in 2015. According to the trend analysis, the habitat quality score presented a linear increasing trend, with a growth rate of 0.004 per year. The increase in habitat quality scores was lower during 2010–2015 and higher during 2000–2005 and 2005–2015. Habitat quality significantly varied among these land uses (Figure 2b). Forestland and grassland had the highest habitat quality score of 0.8. The former increased with a linear growth rate of 0.005 per year, while the latter increased by 0.003 per year. Furthermore, the habitat quality of grassland presented a slight decreasing trend during 2010–2015. The habitat quality score of deforestation was higher than that of afforestation between 2000 and 2010, but lower than that of afforestation in 2015. The habitat quality of deforestation changed with a rate of −0.005 per year during 2000–2015, while that of afforestation increased with the rate of 0.01 per year. The habitat quality of deforestation was lower than that of afforestation during 2010–2015. The habitat quality score of the cropland was approximately 0.55 and increased with the rate of 0.003 per year. In addition, the habitat quality scores of other land use changes were approximately 0.4, but continuously decreased with the rate of −0.009 per year.

### 3.2. Spatial and Temporal Variations of the Habitat Quality

Grids with higher habitat quality in the TGR area were mainly forest landscapes in the northeast and southwest (Figure 3a). The habitat quality score was close to 1 in these regions. The habitat quality of the cropland in the western part of the TGR area is relatively low, with a habitat quality score of about 0.5. Meanwhile, the habitat quality scores of water and build-up land in the western and central of the TGR area were the lowest, which were close to 0. The maximum growth rate of habitat quality score was 0.04 per year and was distributed in the northeastern and southwestern regions (Figure 3b). The maximum reduction rate was −0.05 per year and was mainly located near build-up land and the Yangtze River. In addition, the habitat quality was not significantly improved in the western and northeastern regions.

About 65% of the grids failed to pass the t test at the 95% significant level, which was mainly located in the eastern and central of the reservoir area (Figure 3c). Only 35% of the grids passed the t test at the 95% significant level, which was mainly distributed in the western and northern of the TGR area. The habitat quality change map (Figure 3d) revealed that grids with significantly unchanged habitat quality accounted for 48.5%, which were mainly located in the northeast and western croplands; the grids with a significant increase in habitat quality accounted for 32.47%, which were mainly distributed in the western and central regions of the reservoir area. In addition, only 2.78% of the grids showed a significant decline in habitat quality and were mainly distributed around build-up land and water.

The vertical characteristics of habitat quality in the reservoir area exhibited a distinct spatial heterogeneity (Figure 4). In general, the habitat quality increased with altitude and slope. The habitat quality of >1000 m and >25° zone exceeds 0.8, while the habitat quality of <500 m and <15° zone is less than 0.6. In the altitude zones (Figure 4a), the habitat quality is highest and increased with the rate of 0.0038 per year in the >1000 m zone. The habitat quality is lower in the 500–1000 m zone, but this region had the highest growth rate in habitat quality score of 0.0057 per year. Meanwhile, the habitat quality is lowest in the <500 m zone where had the lowest growth rate of 0.0014 per year. In the slope zones (Figure 4b), the habitat quality is higher and increased at the rate of 0.0053 per year in the 15–25° zone. This change rate is the highest, but the habitat quality of the 15–25° zone was lower than that of the >25° zone. The habitat quality of the >25° zone is the highest and is more than 0.8 in 2000 and reaches 0.9 in 2015. In addition, the habitat quality score of the <15° zone is the lowest and increased at the rate of 0.0021 per year.

### 3.3. Spatial Heterogeneity of the Habitat Quality

The eastern and northern counties have higher habitat quality, while the western counties exhibit lower habitat quality (Figure 5a). The habitat quality is excellent in Xingshan, where the score is about 0.9. The habitat quality scores of four counties (Yiling, Xingshan, Shizhu and Wuxi) exceeded 0.8. Meanwhile, the habitat quality score of Wulong, Fengjie, Yunyang, Wushan, Badong and Zigui is relatively high, and the score fluctuates between 0.7 and 0.8. The other 10 counties in the western part of the reservoir area, such as Jiangjin, Chongqing and Yubei, have habitat quality scores below 0.7. Moreover, the habitat quality is poor in Changshou, where the score is only 0.52. According to the trend analysis and significance test, the habitat quality of all the counties improved except for Yubei and Chongqing (Figure 5b). The habitat quality of 7 counties significantly changed, such as Zigui, Wuxi and Kai County. Among them, the habitat quality of Chongqing has significantly deteriorated, where its habitat quality score decreased with a value of 0.002 per year. The habitat quality scores of Wuxi, Zigui and Jiangjin increased with a value of 0.004 per year. Moreover, the habitat quality of other 13 counties did not pass the significance test. Among these counties, the largest increment of habitat quality was found in Yunyang and Wushan, with a value of 0.008 per year.

The spatial distribution and dynamics of the habitat quality in these counties could be divided into 3 categories (Figure 6). First, counties with a high level of habitat quality. There are six counties such as Xingshan, Yiling and Badong, which were mainly located in the eastern part of the reservoir area. Their habitat quality scores were up to 0.9. Second, counties with a middle level of habitat quality. There are counties such as Yunyang, Fengjie and Wushan in the central part of the reservoir area, whose habitat quality scores were about 0.7. Third, counties with a low level of habitat quality. There were three counties, including Changshou, Zhongxian and Yubei, and Chongqing City. They were distributed in the western part of the reservoir area and their habitat quality scores were about 0.5 between 2000 and 2015.

### 3.4. Relationship between Landscape Dynamic and Habitat Quality

The first dimension could explain 70.2% of the changes in landscape dynamics and habitat quality, while the second dimension explain only 29.8%. Most of the landscape change categories were concentrated around the no change in habitat quality (Hab) (Figure 7), which indicated that these landscape dynamics could not significantly alter habitat quality. Afforestation (F+) and the loss of cropland (C−) were distributed around the increase in habitat quality (Hab+). It indicated these landscape dynamics could effectively enhance habitat quality. The decrease in habitat quality (Hab−) and the increase in the built-up land (B+) are close together, showing the urbanization and urban expansion would significantly reduce habitat quality. Moreover, G+, W+ and B− were distributed between Hab and Hab−, indicating these dynamics may also cause deterioration of habitat quality.

## 4. Discussion

### 4.1. Spatial and Temporal Variations of the Habitat Quality

The habitat quality of the TGR area was assessed by the InVEST model and presented a significant increase trend (Figure 2a). This result supported the notion that human activities have effectively improved the habitat quality of the reservoir area, which is consistent with the other studies such as Chu et al. [13]. In addition, the habitat quality and its changes presented a significant spatial heterogeneity (Figure 3). The habitat quality was higher in the northeastern and southwestern parts of the reservoir area, whose main land use type was forest. The forest ecosystem has a complex structure and contains a self-recovery ability when facing disturbances from human activities and can maintain higher habitat quality. This was consistent with the findings of Bai et al. [15], who found that human impacts on natural habitats were relatively weak and therefore the habitat quality in these areas was better maintained. The habitat quality was not improved in the western croplands and the northeastern regions, while habitat quality significantly improved in the central regions. The main reason for this difference was the implementation of these ecological projects such as the Three North Shelterbelt Project and the Natural Forest Protection Project in the central regions [19,28,29]. The lowest habitat quality score was mainly located in the regions near water and western build-up land which presented a clear deterioration trend of habitat quality (Figure 3c). This was mainly due to the fact that urban and rural settlements and cropland were distributed around water. And the claiming of natural resources by human activities and the discharge of pollutants have caused damage to regional ecosystems, resulting in a sharp decline in habitat quality. In addition, urbanization and urban expansion have exacerbated habitat fragmentation, rendering the connectivity between habitat patches poorer and ultimately leading to the deterioration of the surrounding habitat quality [25].

The habitat quality scores increased with altitude and slope and showed an increasing trend over time in the altitude and slope zones (Figure 4). The main reason is that intense human activities usually occur in the regions of <500 m and <15°. Serving as an important habitat and foraging ground for mammals and birds [5,12,41], the configuration of vegetation affects the spatial heterogeneity of habitat quality at the regional scale. Natural vegetation damage and vegetation recovery caused by human activities are key factors of habitat quality dynamics [19,42]. Since the ecological demands of different counties are determined by many environment factors (such as topography, soil, and climate), the impact of human activities on habitat quality is also unavoidable. Therefore, building ecological buffer zones can effectively improve habitat quality and protect regional ecological environments. Furthermore, vegetation restoration in protected areas should focus on the impact of different landscapes on the regional biodiversity and habitat quality.

### 4.2. Response of Habitat Quality on Landscape Dynamics

Habitat quality and its changes significantly varied among vegetation landscapes (Figure 2b). By comparing the distance among these landscape changes and the habitat quality changes in the corresponding analysis diagram (Figure 7), both afforestation and the loss of cropland could effectively increase habitat quality. However, urbanization and urban expansion, characterized by an increase in built-up land, resulted in a significant reduction in habitat quality [43]. This suggested that habitat quality was extremely sensitive to the response of vegetation recovery and urban expansion, and human activity disturbance from a single behavioral model could directly alter the habitat quality of a certain location [19,44]. In addition, theoretical and empirical studies have identified a variety of linkages between restoration actions and biodiversity changes [45]. Ecological restoration on degraded forestland or grassland may not significantly increase biodiversity, but revegetation on the cropland would significantly alter biodiversity. Therefore, biodiversity conservation should more focus on the revegetation of cropland. The focus is on comparing the applicability of active and passive restoration methods in different habitats to select the more effective methods for biodiversity restoration [19].

Improving forest landscape configuration could contribute to the overall enhancement of ecosystem services [46,47], which has a great significance for the maintenance and restoration of biodiversity [42]. Afforestation activities would add new patches to the forestland or expand outwards along the original landscape, causing the changes in the mean patch area and its coefficient of variation show synchrony in time. Due to the coupling influence of forestry ecological projects, the TGD, and urbanization, the fragmentation gradually decreased, and the aggregation increased for the forestland [28,33]. Meanwhile, cropland significantly decreased, which exacerbated the landscape fragmentation and reduced the aggregation of agricultural landscapes. Although severe fragmentation of agricultural landscapes could reduce environmental risks such as regional soil erosion and non-point source pollution [48], it may pose a threat to the livelihoods of farming populations. In addition, the fragmentation of built-up land has increased due to urban expansion scattered on counties, as well as the coefficient of variation of built-up land patches has significantly increased, indicating a gradual increase in the variation in urban scale among counties. Some built-up land changes may be associated with ecological migration from the reservoir area in the pre-storage period and large-scale agricultural population movements to large cities in the post-storage period [49]. Therefore, it is necessary to pay attention to the optimization and adjustment of the land use structure in the ecological migration areas, and to ensure the sustainable development of the migration areas in order to avoid further negative impacts on the ecological environments.

### 4.3. Enlightenments of Biodiversity Conservation

Biodiversity monitoring and assessment provided the scientific basis for ecosystem restoration and biodiversity conservation [6,7]. In past decades, vegetation restoration in the TGR area neglected the biodiversity differentiation of complex community structures, while enhanced the biodiversity on quantity by the means of replacing croplands with a single community [13,45]. This behavior leads to a relatively simple structure and species composition of the restocked ecosystem. With the gradual reduction in sloping cropland, it is unreasonable to regulate the landscape by means of returning cropland to forest. Therefore, future vegetation landscape regulation should aim to build a complex vertically distributed community structure. In addition, multiple species used in the restoration were proved to improve multiple and stable ecosystem functions [50], so we need some appropriate pioneer species and species diversity to support long-term ecosystem recovery and succession [51].

Conflict between biodiversity conservation and economic development is a pressing challenge for restoration ecology. Now, combining ecosystem service supply and human survival needs based on the spatial heterogeneity of the ecological environment to regulate landscape is becoming the prevailing view of restoration ecology [12]. As a typical ecological fragile area in China, the TGR area has been strongly disturbed by human activities, leading to a series of serious environmental problems. Therefore, landscape regulation of the TGR area in future should take full account of its spatial heterogeneity of ecological environment and economic construction to subzone the reservoir area’s habitat quality rationally, and finally execute the recovery targets according to various demands from different zones. At present, some ecological recovery projects on the reservoir area are being carried out, such as returning farmland to forest and Yangtze River Protection Forest Project [33]. Although they have effectively increased the forest area of the reservoir area, most areas still suffered from many problems such as low coverage and single species [19]. In addition, as the ability of supporting regional sustainable development varies among different land use types, the spatial configuration of landscape will affect the regional habitat quality. Therefore, ecological protection should construct rational configuration of various landscape types. A decision-maker should also focus on protecting ecological source sites such as forestlands and grasslands, and reasonably delineate ecological protection red lines, thereby reducing the damage to ecological sources and ecological corridors caused by urbanization and urban expansion. The purpose is to improve the connectivity between habitat patches, and ultimately improve the overall habitat quality of the reservoir area [52].

### 4.4. Landscape Planning for Recovering Habitat Quality

According to the theoretical research results, landscape dynamics such as afforestation could alter the habitat quality. The TGR area has three typical habitat regions, i.e., ecological protection areas, agroforestry areas, ecological barriers and buffer zones around construction lands and water bodies. Moreover, the island of the reservoir area is also a unique habitat. Therefore, we proposed some landscape planning suggestions (Figure 8) for four typical regions to recover habitat quality. These suggestions also are applicable to the ecological restoration of other ecologically fragile areas or areas with similar ecosystem characteristics.

Ecological protection areas mainly refer to the northeast and high-altitude areas of the TGR area and are covered by forests and grasses (Figure 8a). Large vegetation patches not only provide habitats for animals but are also important foraging areas [19]. We suggest increasing large patches by reducing landscape fragmentation and increasing landscape connectivity. Moreover, stable biological communities should be built in these vegetation patches, and various species should be introduced to optimize the vegetation configuration and the community vertical structure.

Agroforestry areas mainly refer to the middle of the TGR area and are covered by forests and crops (Figure 8b). Returning farmland to forests is the most important human activity in this region in the past few decades. However, monotonous crop and tree types are difficult to provide better habitat conditions [19]. We suggested organically integrating various forest types such as protection forests, timber forests and economic forests, and build complex agroforestry network with integrating tree, shrub, grass and crop. Meanwhile, landscape planning should adopt high standard agroforestry structure design, and focus on the stability of the agroforestry ecosystem to improve comprehensively the habitat quality and the food supply capacity of this region.

Ecological barriers and buffer zones are mainly distributed in the southwest and low-altitude areas of the reservoir area (Figure 8c). Cultivation of crops, grazing and construction of buildings in this zone are discouraged. It is better to establish a multi-species vegetation belt with an integration among trees, shrubs, and herbs, which could increase the habitat quality by beautifying the waterscapes and purifying pollutants generated by industry and life along the reservoir shore. Meanwhile, economic forests can be properly developed in the suitable ecological areas, which could not only increase the diversities of ecosystem and landscape, but also promote local economic development. Moreover, riverfront parks could be proposed and constructed at the top of the flood protection embankment to beautify the surrounding urban landscape and increase the habitat qualities by changing the hard embankments into ecological embankments.

The islands in the reservoir area can be divided into two categories, i.e., small islands at low altitudes that may be submerged and large islands at high altitudes that will not be submerged. The former should be established as nature protection areas and strictly protected, while the latter could develop some ecological constructions and activities. We proposed some landscape planning suggestions for the latter islands (Figure 8d). Ecological restorations (such as restoring the mountain, planting trees and forests, and replanting native herbs on the isolated island) and ecological constructions (such as building an ecological walking system around the island) should be carried out at the same time. The exposed cliffs and slopes should be repaired through topographical finishing to build a three-dimensional landscape. Color-leaf tree species and flowering tree species could be planted on the slopes to build a complex forest landscape, which could enhance the vegetation succession ability and biodiversity. Meanwhile, native long-lived vegetation and original tree species are used for reforestation to improve the ecosystem stability and the habitat quality.

### 4.5. Limitations and Uncertainties

Biodiversity is an important ecosystem characteristic. However, there exists a dispute regarding the relationship between biodiversity and ecosystem service [6,7]. One view is that biodiversity underpins ecosystem services, while another view maintains that biodiversity is a supporting service within ecosystem services. Due to the unclear definition of their relationship, we completed the reforestation activities with a relatively single tree species in the early stages of ecological restoration, which has seriously affected the vegetation evolution process [53]. This paper used the habitat quality to measure biodiversity levels, and we took it as an important indicator of biodiversity conservation and restoration. The InVEST model could effectively respond to differences in habitat quality in physical environments such as topography, soils, landscape types and threats. However, its results were assessed mainly on the basis of surface landscape and fail to take into account differences in biotypes within the same landscape, which was the limitation of habitat quality assessment at a regional scale. Future studies could consider quadrant measurements of species in the same landscape using multiple landscape habitat quality parameters such as Shannon’s diversity index, Shannon’s evenness index, the largest patch index, and so on.

Threats in the model are weighted and summed to affect habitats in the landscape, while it is difficult to simulate the interaction of multiple threats, which is limited by the mechanism of biological responses to complex environments [6,42]. Currently, numerous sample experiments research of biodiversity response to coupled factors were carried out [7], but the results were different to embed in ecological models. In addition, complex model parameters are an important reason why mechanistic ecological models are difficult to apply to a wide range of studies [54]. Although the InVEST model can evaluate many service functions of ecosystems including biodiversity with advantages of less input data, low application cost and easy to operate [13], the model evaluation accuracy needs to be improved. However, it effectively reflects the spatial heterogeneity of habitat quality, which is an advantage unmatched by small-scale biodiversity conservation studies [19]. Therefore, there still need further improvements and consideration in the setting of model parameters and the selection of threat factors.

## 5. Conclusions

The habitat value has been widely recognized from both social and ecological perspectives, as habitat is a key element for the continued survival of species and can provide multiple ecosystem services to humans. However, there is a lack of studies revealing the role of land use change in habitat degradation from a multi-scale perspective. Therefore, taking the TGR area as the study area, we assessed the spatial and temporal variation of habitat quality in this area using the habitat quality module of the InVEST model, and analyzed the response of habitat quality to landscape dynamics from different perspectives. The results demonstrated that the habitat quality scores of terrestrial ecosystems of the reservoir area increased linearly over time and presented a significant spatial heterogeneity. Habitat quality was significantly higher in the northeastern and southwestern parts of the reservoir area, and has remarkably improved in the central part, while it has significantly deteriorated in the southwest around built-up land and water. Furthermore, the habitat quality score increased with altitude and slope, and showed an upward trend over time. In addition, habitat quality and its changes significantly vary between vegetated landscapes. The results demonstrated that afforestation and the loss of cropland could effectively increase the level of the habitat quality while urban expansion could decrease it. Moreover, the habitat quality of single landscapes at the regional scale was consistently increasing due to the combined effects of ecological restoration and landscape connectivity. Although the habitat quality in the TGR area has significantly improved, the severe human-land conflict has placed new demands on biodiversity conservation and restoration. Therefore, we suggested that the future landscape regulation of the TGR area should take full account of its spatial heterogeneity of ecological environment and economic construction to subzone the reservoir area’s habitat quality rationally, thereby execute the recovery targets according to different needs from different zones. Our study could provide a theoretical basis for biodiversity conservation and the assessment and management of ecosystem health, as well as guide the management and decision-making of landscapes at the regional scale.

## Figures and Tables

**Figure 1 ijerph-19-03594-f001:**
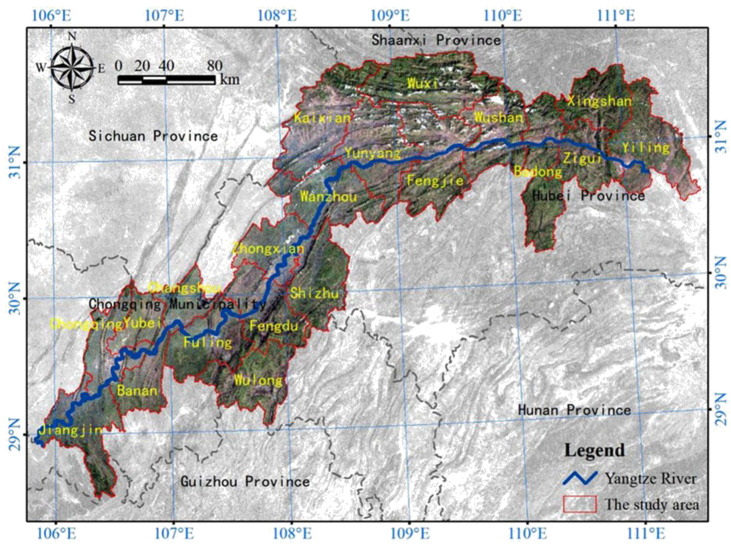
Location of the Three Gorges Reservoir area in China.

**Figure 2 ijerph-19-03594-f002:**
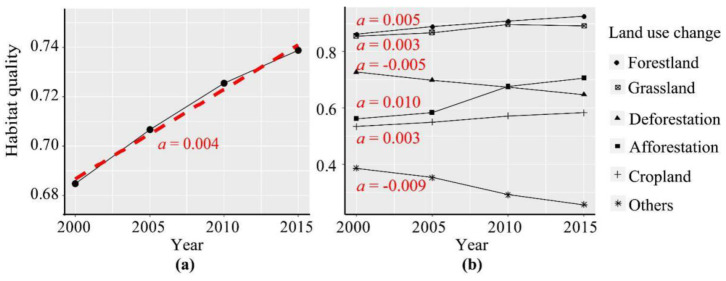
Linear trend analysis of the habitat quality of the TGR area (**a**) and temporal variations of the habitat quality for different land use change types (**b**). Note: In order to quantify the dynamic of habitat quality score, we used a least-square linear regression model to fit the habitat quality score. The changing trend is described by the modelled slope which is a in the figure. The black broken line is the estimated results, and the red dotted line is the trend line of habitat quality score.

**Figure 3 ijerph-19-03594-f003:**
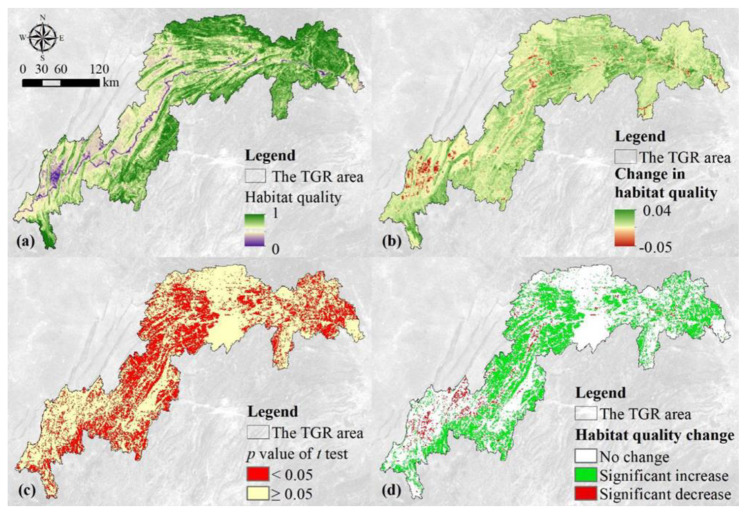
Spatio-temporal variation of the habitat quality in the TGR area between 2000 and 2015. Average (**a**) and the modelled slope (**b**) of the habitat quality, and *p* value of t test for the modelled slope (**c**) and the habitat quality changes (**d**). Note: In the figure (**d**), no change documents that the habitat quality did significantly no change (slope = 0), and significant increase refers to slope > 0 and *p* < 0.05, while significant decrease refers to slope < 0 and *p* < 0.05.

**Figure 4 ijerph-19-03594-f004:**
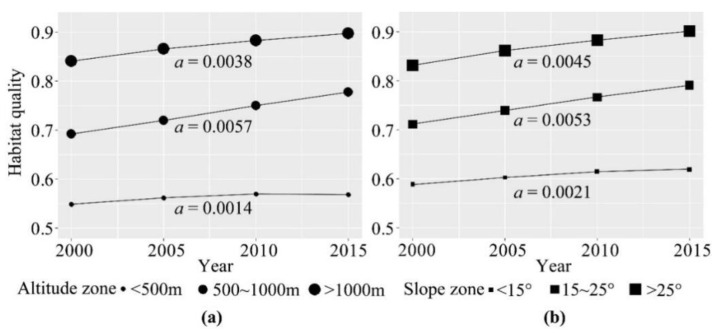
Temporal variations of the habitat quality in different altitude zones (**a**) and slope zones (**b**) in the TGR area. Note: We used a least-square linear regression model to fit the habitat quality score. The changing trend is described by the modelled slope which is a in the figure.

**Figure 5 ijerph-19-03594-f005:**
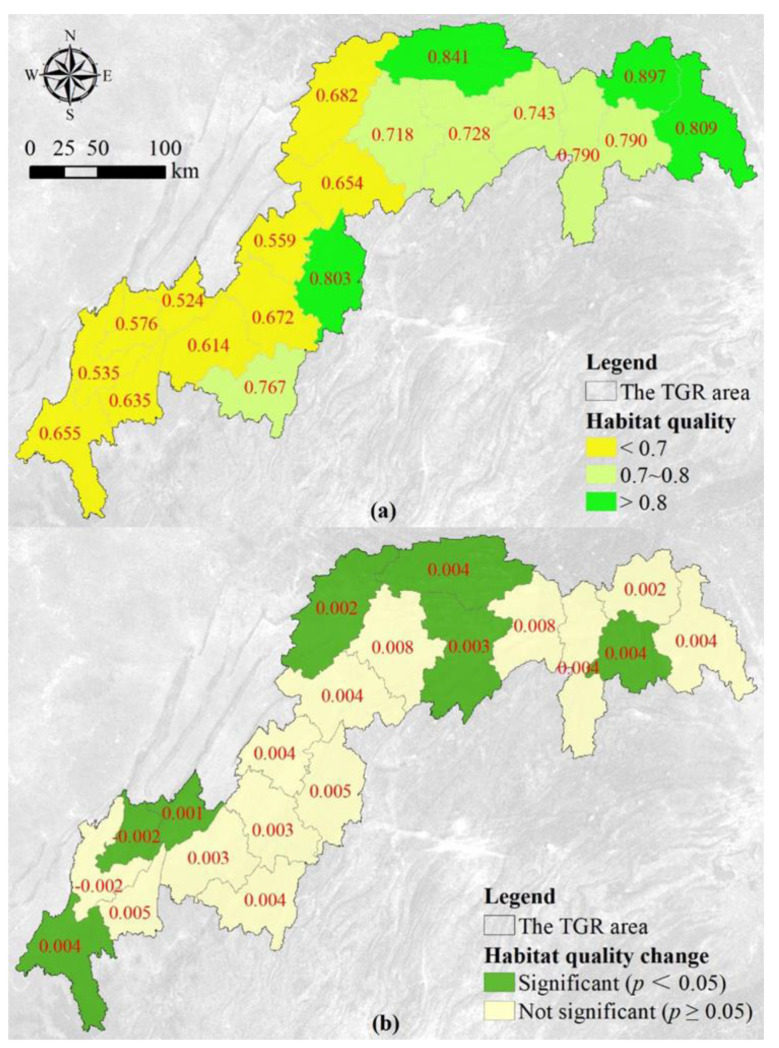
Average (**a**) and the modelled slope (**b**) of the habitat quality between 2000 and 2015 at county scale.

**Figure 6 ijerph-19-03594-f006:**
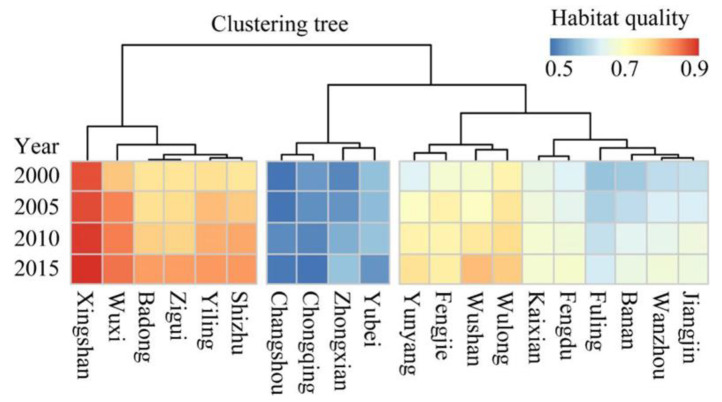
Heat map of correlation matrix and hierarchical clustering for 20 counties in the TGR area according to habitat quality.

**Figure 7 ijerph-19-03594-f007:**
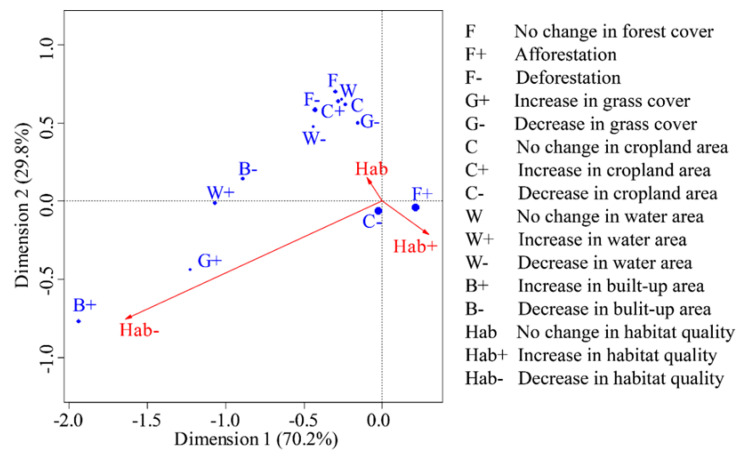
Correspondence analysis between habitat quality changes and land use changes.

**Figure 8 ijerph-19-03594-f008:**
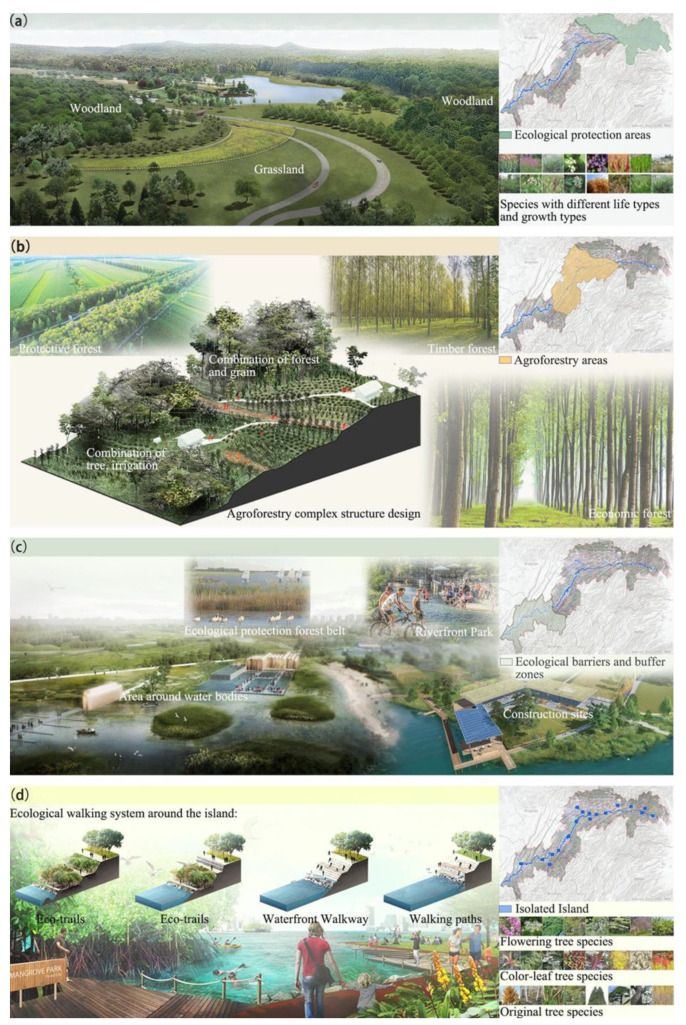
Landscape planning for ecological protection areas (**a**), agroforestry areas (**b**), ecological barriers and buffer zones (**c**), and large islands at high altitudes (**d**) to recover habitat quality.

**Table 1 ijerph-19-03594-t001:** The study data and description.

Data	Data Sources	Description
Land use data	30 m resolution land use maps were derived from Huang et al. [29].	Land use maps with nine land use types (coniferous forest, broadleaf forest, mixed forest, shrub, grassland, cropland, water, built-up land, and bare land) of the TGR area in 2000, 2005, 2010 and 2015.
DEM data	30 m resolution DEM data were derived from ASTER Global Digital Elevation Model V002 (http://www.gscloud.cn/, accessed on 13 November 2018).	DEM was used to identify the elevation and slope of the TGR area, and to generate altitude zones and slope zones.
Railway data	National Geomatics Center of China (NGCC) (http://ngcc.sbsm.gov.cn/, accessed on 13 November 2018).	Railways within the TGR area in 2000, 2005, 2010 and 2015.
Highway data	National Geomatics Center of China (NGCC) (http://ngcc.sbsm.gov.cn/, accessed on 13 November 2018).	Highways within the TGR area in 2000, 2005, 2010 and 2015.
National road data	National Geomatics Center of China (NGCC) (http://ngcc.sbsm.gov.cn/, accessed on 13 November 2018).	National roads within the TGR area in 2000, 2005, 2010 and 2015.
Traffic station data	Points of tourist attractions were acquired from Baidu maps.	Traffic stations within the TGR area in 2000, 2005, 2010 and 2015.
Hotel data	Points of hotels were acquired from Baidu maps.	Hotels within the TGR area in 2000, 2005, 2010 and 2015.
Tourist attraction data	Points of traffic stations were acquired from Baidu maps.	Tourist attractions within the TGR area in 2000, 2005, 2010 and 2015.

**Table 2 ijerph-19-03594-t002:** Threat factors of habitat quality and their attributes.

Threat Factor	Data Type	Maximum Distance	Decay Type	Weight
Railway	Linear vector data	1 km	Exponential distance-decay function	0.5
Highway	Linear vector data	2 km	Exponential distance-decay function	0.8
National road	Linear vector data	1 km	Exponential distance-decay function	0.8
Traffic station	Point vector data	10 km	Linear distance-decay function	1
Hotel	Point vector data	5 km	Linear distance-decay function	0.7
Tourist attraction	Point vector data	3 km	Linear distance-decay function	0.6
Built-up area	Raster data	10 km	Exponential distance-decay function	1
Water	Raster data	1 km	Linear distance-decay function	0.3
Cropland	Raster data	4 km	Linear distance-decay function	0.5

**Table 3 ijerph-19-03594-t003:** Habitat scores and the responses to threat factors of land use types.

Land Use Type	Habitat Score	The Relative Sensitivity of Each Habitat Type to Each Threat								
		Railway	Highway	National Road	Hotel	Traffic Station	Tourist Attraction	Built-up Area	Water	Crop Land
Coniferous forest	1	0.6	0.7	0.8	0.8	0.9	0.5	0.9	0.8	0.5
Broadleaf forest	1	0.6	0.7	0.8	0.8	0.9	0.5	0.9	0.8	0.5
Mixed forest	1	0.6	0.7	0.8	0.8	0.9	0.5	0.9	0.8	0.5
Shrub	1	0.3	0.3	0.3	0.3	0.3	0.3	0.4	0.5	0.75
Grassland	0.8	0.6	0.7	0.8	0.5	0.5	0.5	0.5	0.5	0.75
Cropland	0.5	0.6	0.8	0.8	0.5	0.9	0.5	1	0.8	0.75
Water	0	0	0	0	0	0	0	0	0	0
Built-up land	0	0	0	0	0	0	0	0	0	0
Bare land	0	0	0	0	0	0	0	0	0	0

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
