# Peer review of "Spatial and Temporal Variations of Habitat Quality and Its Response of Landscape Dynamic in the Three Gorges Reservoir Area, China"

_ijerph, 2022, doi:10.3390/ijerph19063594_

Round 1

Reviewer 1 Report

Review Report - Spatial and temporal variations of habitat quality and its response of landscape dynamic in the Three Gorges Reservoir area, China

Comments to authors

The authors aimed to understand the relationship between habitat loss and land use change over time, using a case study of the Three Gorges Reservoir. 

To address this, they analyzed temporal and spatial changes using the InVEST habitat quality model, and created summary quality scores for each habitat region. This is an active research area and would be interesting to readers. 

I enjoyed reading this article. Your work is valuable and can be better highlighted by (1) a careful review of the grammar and phrasing of the manuscript (2) modifications and suggested additions to the manuscript, especially around the introduction. 

Specific comments:

  1. Please review manuscript. Many areas of the manuscript could be reviewed for phrasing. For example, on Line 198: “was determined to grasp the different damage potentials of each threat” is unclear what you mean. Do you mean the variable is attempting to capture these damage potentials? If so, please state that. Please have this manuscript reviewed carefully, as we want to ensure the ideas are clear and concise for readers. 
  2. On Line 95, you mention the habitat quality module of the InVEST model, but it would be useful to compare it to other methods to estimate habitat quality before you focus on this one. I suggest you provide a few more citations. I like your introduction overall. 
  3. Figure 1: The font in the map is a bit small, can you increase the study area labels slightly? Also increasing the red line thickness would be useful for those (like myself) who are hard at sight.
  4. Line 321: Which package and function did you use for correspondence analysis? Please state this and provide all the details needed to reproduce this, or provide the source code. 
  5. Figure 2: Is it possible to put uncertainty measures on your point estimates? e.g. error bars

Author Response

Response to Reviewer #1:

The authors aimed to understand the relationship between habitat loss and land use change over time, using a case study of the Three Gorges Reservoir.

To address this, they analyzed temporal and spatial changes using the InVEST habitat quality model, and created summary quality scores for each habitat region. This is an active research area and would be interesting to readers.

I enjoyed reading this article. Your work is valuable and can be better highlighted by (1) a careful review of the grammar and phrasing of the manuscript (2) modifications and suggested additions to the manuscript, especially around the introduction.

Comment 1: Please review manuscript. Many areas of the manuscript could be reviewed for phrasing. For example, on Line 198: “was determined to grasp the different damage potentials of each threat” is unclear what you mean. Do you mean the variable is attempting to capture these damage potentials? If so, please state that. Please have this manuscript reviewed carefully, as we want to ensure the ideas are clear and concise for readers.

Response 1: Thank you for your suggestion. We have reviewed our manuscript carefully. Moreover, for the problem on line 200, “some threats may be more damaging to habitat, all else equal, and a relative impact score accounts for this” could be found in the InVEST User’s Guide to interpret “the relative impact of each threat”. To avoid ambiguity, the “grasp” was replaced by “represent”.

Reference:

Sharp, R.; Chaplin-Kramer, R.; Wood, S.; Guerry, A.; Tallis, H.; Ricketts, T.; Nelson, E.; Ennaanay, D.; Wolny, S.; Olwero, N.; et al. InVEST User’s Guide, 2018. http://dx.doi.org/10.13140/RG.2.2.32693.78567

Comment 2: On Line 95, you mention the habitat quality module of the InVEST model, but it would be useful to compare it to other methods to estimate habitat quality before you focus on this one. I suggest you provide a few more citations. I like your introduction overall.

Response 2: We’re honored to receive your appreciation for the introductory section and thank you for your suggestion. According to your suggestion, we added information on two additional ecological models (i.e., ARIES and SoIVES) and pointed out the shortcomings of their application to existing habitat quality assessments, which could be found in lines 91-95.

Comment 3: Figure 1: The font in the map is a bit small, can you increase the study area labels slightly? Also increasing the red line thickness would be useful for those (like myself) who are hard at sight.

Response 3: We're sorry for the trouble you've caused reading, and based on your suggestions, we've revised Figure 1.

Comment 4: Line 321: Which package and function did you use for correspondence analysis? Please state this and provide all the details needed to reproduce this, or provide the source code.

Response 4: Thanks for to your comment. We used ‘ca’ function in R language for correspondence analysis, and a more detailed procedure for the corresponding analysis can be found in lines 318-327. 

Comment 5: Figure 2: Is it possible to put uncertainty measures on your point estimates? e.g. error bars

Response 5: Thanks for to your suggestion. Figure 2 depicts the trends in habitat quality over time for different land use types. We have verified the R2 and RSME of the model when performing the trend analysis, thus, we have not represented the uncertainty measures for the overall presentation.

Reviewer 2 Report

The English of this paper should be polished by native English speakers.

Line 36-41, the argument should be supported by references. Otherwise, it will become arbitrary.

Line 47-48, this sentence is meaningless here as it presents the same meaning as the one in line 45-47.

Line 57-60, I am not convinced by your argument even you have provided references. Moreover, it is not logic with the contents in line 61-63.

Line 74-75, I do not know why authors shift their attention to scale.

Line 93-94, this sentence is not needed here. What I want to know is its shortcomings.

After line 101, I cannot catch a general research gap that is suitable for all scholars and then the TGR can be adopted as a case study to address the general research gap. (This is important for a high-quality research paper).

Please move line 102-127 to study area or methods.

We are now in the year of 2022, so that I think it is more proper to include the data of 2020. The data of 2015 is too old.

Line 303-312, I do not think the title of trend analysis is proper. Trend analysis cannot be a method. Least square linear regression analysis.

Figure 2, authors have not provided the R2, RSME to assess the model.

Figure 4, authors have not provided the R2, RSME to assess the model.

Line 431, I do not think Chongqing is a county.

Figure 8, I do not think Fig.8 can be properly described in Line 567-607. I suggest authors should remove Fig.8. It is also wise to enhance the discussion logically.

Author Response

Response to Reviewer #2:

Comment 1: Line 36-41, the argument should be supported by references. Otherwise, it will become arbitrary.

Response 1: Thanks for your comments. We have added the appropriate references in our manuscript.

Comment 2: Line 47-48, this sentence is meaningless here as it presents the same meaning as the one in line 45-47.

Response 2: Thank you for your suggestion. Based on your suggestion, we deleted the sentence in lines 48-49.

Comment 3: Line 57-60, I am not convinced by your argument even you have provided references. Moreover, it is not logic with the contents in line 61-63.

Response 3: Thank you for your comments. Based on your comments, we have revised the manuscript as detailed in lines 59-65.

Comment 4: Line 74-75, I do not know why authors shift their attention to scale.

Response 4: We're sorry that our ideographic ambiguity has led you to think that we're turning our attention to the study of multiple scales, and what we were actually trying to say was that there is a lack in understanding the role of LULC change on habitat. The corresponding part of the manuscript has been revised, which could be found in lines 82-84.

Comment 5: Line 93-94, this sentence is not needed here. What I want to know is its shortcomings.

Response 5: Thank you for your suggestion. According to your suggestion, we added information on two additional ecological models (i.e., ARIES and SoIVES) and pointed out the shortcomings of their application to existing habitat quality assessments, which could be found in lines 91-95.

Comment 6: After line 101, I cannot catch a general research gap that is suitable for all scholars and then the TGR can be adopted as a case study to address the general research gap. (This is important for a high-quality research paper).

Response 6: Thanks for your comments. First, our study fills such a research gap by quantifying the impact of landscape pattern evolution on habitat quality in ecologically fragile areas under the implementation of ecological restoration projects. Second, although the study results cannot catch a general research gap that is suitable for all scholars, they can provide some academic reference value for the implementation of ecological restoration projects and habitat conservation work in other similar areas.

Comment 7: Please move lines 102-127 to study area or methods.

Response 7: Thank you for your comment. In this paragraph, we present the social context in which the Three Gorges reservoir area has evolved into an ecologically fragile area due to the significant impact of economic construction and urban expansion on the ecological environment. In addition, the impact of the implementation of a large number of ecological restoration projects on the biodiversity of the study area is unclear, which is also the purpose of our study. After discussion, we felt that it was necessary to introduce the information clearly before the main study. Therefore, we believe that this paragraph is justified in this location.

Comment 8: We are now in the year of 2022, so that I think it is more proper to include the data of 2020. The data of 2015 is too old.

Response 8: Thank you for your suggestions. The period 2000-2015 was chosen as the study time period for several reasons. First, the construction of the 180-m-tall dam was officially started in 1994 and the water level rise to 175 m after the three Gorges Reservoir impoundment in 2009 (Chu et al., 2018). The effects of dam construction on changing landscape patterns of biological habitats have become evident after 2010. Second, a number of national ecological restoration projects began in 1998 (Teng et al., 2019), and the effect of their implementation on the evolution of the landscape pattern has been demonstrated. In the context of the above study, we believe that this research paragraph can show the effect of the evolution of landscape pattern on habitat quality under the effect of economic development and ecological restoration.

Reference:

Teng M, Huang C, Wang P, et al. Impacts of forest restoration on soil erosion in the Three Gorges Reservoir area, China. Science of The Total Environment, 2019, 697:134164.

Chu L, Sun T, Wang T, et al. Evolution and Prediction of Landscape Pattern and Habitat Quality Based on CA-Markov and InVEST Model in Hubei Section of Three Gorges Reservoir Area (TGRA). Sustainability, 2018, 11:3854.

Comment 9: Line 303-312, I do not think the title of trend analysis is proper. Trend analysis cannot be a method. Least square linear regression analysis.

Response 9: Thanks for your suggestions. According to the relevant reference, trend analysis uses mathematical and statistical techniques to extend time series data into the future (Wu et al., 2011). Meanwhile, we mainly used this method to identify the changing trend of habitat quality, while the least squares regression method aided in analyzing this trend by modeling. Therefore, we think that the trend analysis method is justified under the heading of statistical analysis methods.

Reference:

Feng-Shang Wu, Chun-Chi Hsu, Pei-Chun Lee, et al. A systematic approach for integrated trend analysis—The case of etching. Technological Forecasting and Social Change, 2011, 78(3):386-407.

Comment 10: Figure 2, authors have not provided the R2, RSME to assess the model.

Response 10: Thanks for to your suggestion. Figure 2 depicts the trends in habitat quality over time for different land use types. We have verified the R2 and RSME of the model when performing the trend analysis, and we did not present it for the sake of the overall presentation effect.

Comment 11: Figure 4, authors have not provided the R2, RSME to assess the model.

Response 11: Thank you for to your comment. As mentioned above, we have verified the R2 and RSME of the model when performing the trend analysis, and we did not present it for the sake of the overall presentation effect.

Comment 12: Line 431, I do not think Chongqing is a county.

Response 12: Thanks for your comment. We have listed the city of Chongqing separately rather than including it in counties, which could be found in lines 443-444.

Comment 13: Figure 8, I do not think Fig.8 can be properly described in Line 567-607. I suggest authors should remove Fig.8. It is also wise to enhance the discussion logically.

Response 13: Thanks for your comment. We decided to keep the Fig.8 after discussion. The reasons are as follows. The specific landscape planning recommendations we made are comprehensive, covering different landscape types, and implementable. It is also the practical value of our study. And Figure 8 is a visual representation of our planning proposal, which is helpful for the reader to understand our scheme better.

Reviewer 3 Report

Dear Authors,

I like your article. In my opinion your manuscript is generally well-written and structured. It is  legible and understandable. Regarding its scientific soundness  the methodology can be discussed, but it is consistent and honest, results are useful and interesting. The background of literature looks adequate to these studies. The research is well done and the discussion is very well presented where limitations of the study are clearly described and show possible ways of developing the research further, in the future.

Author Response

Response to Reviewer #3:

Dear Authors,

I like your article. In my opinion your manuscript is generally well-written and structured. It is legible and understandable. Regarding its scientific soundness the methodology can be discussed, but it is consistent and honest, results are useful and interesting. The background of literature looks adequate to these studies. The research is well done and the discussion is very well presented where limitations of the study are clearly described and show possible ways of developing the research further, in the future.

Response:We are very grateful to the reviewer for the valuable comments on our manuscript. Our research analyzed the spatial and temporal variations of habitat quality of the Three Gorges Reservoir area by the InVEST habitat quality model, and demonstrated the responses of habitat quality to various landscape dynamics by correspondence analysis. The aim is to provide a scientific basis for formulating regional ecological conservation policies and sustainable use of land resources.

Reviewer 4 Report

Dear Authors,

The paper is prepared very well, I have just a few comments, which I write line by line:

31 - keywords with small first letters

43 - [2,3,4,5], whereas it should be like [2-5]. The same in line no. 50.

109 - always put a space between number and e.g. km, so 38 km. The same problem in 149 line. 

140 -always put a space between Fig. and number like here: (Fig. 1).

162-163 - m instead of metres. 1300 m (with space).

207-208 - (Eq. 1) instead of (Eq.1). 229, etc- the same thing.

256 - first record of the table: 30 m

274 - put space between number and km

339 - (Fig. 2a) instead of (Fig.2 a). the same thing in other places in the text.

382 - Spatio-temporal 

485 - do not start a setence with "And....".

669 - Author contributions:  put only initials of name and surname

681-682 - do not use a big letters: Wildlife Society Bulletin

683 - something is wrong inside name of journal

695, 699, 704, 740, 773, etc. - Everywhere in the references you should put short names of journals. Moreover, here DOI is not as a link (blue one like almost everywhere). This mistake is repeated many times.

711, 727 - without Vol

745 -  southwest china normal university press  - probably it should be with first big letter, because it is name, am I right?

747 - lack of short name and small first letters.

Author Response

(The authors gave the same response as above.)

Round 2

Reviewer 2 Report

well done